# On the Fairness of Disentangled Representations

Francesco Locatello[2,5], Gabriele Abbati[3], Tom Rainforth[4], Stefan Bauer[5], Bernhard Schölkopf[5], and Olivier Bachem[1]

[1]Google Research, Brain Team
[2]Dept. of Computer Science, ETH Zurich
[3]Dept. of Engineering Science, University of Oxford
[4]Dept. of Statistics, University of Oxford
[5]Max-Planck Institute for Intelligent Systems

## Abstract

Recently there has been a significant interest in learning disentangled representations, as they promise increased interpretability, generalization to unseen scenarios and faster learning on downstream tasks. In this paper, we investigate the usefulness of different notions of disentanglement for improving the fairness of downstream prediction tasks based on representations. We consider the setting where the goal is to predict a target variable based on the learned representation of high-dimensional observations (such as images) that depend on both the target variable and an *unobserved* sensitive variable. We show that in this setting both the optimal and empirical predictions can be unfair, even if the target variable and the sensitive variable are independent. Analyzing the representations of more than $12\,600$ trained state-of-the-art disentangled models, we observe that several disentanglement scores are consistently correlated with increased fairness, suggesting that disentanglement may be a useful property to encourage fairness when sensitive variables are not observed.

## 1 Introduction

In representation learning, observations are often assumed to be samples from a random variable $\mathbf{x}$ which is generated by a set of unobserved factors of variation $\mathbf{z}$ [6, 14, 53, 89]. Informally, the goal of representation learning is to find a transformation $r(\mathbf{x})$ of the data which is useful for different downstream classification tasks [6]. A recent line of work argues that disentangled representations offer many of the desired properties of useful representations. Indeed, isolating each independent factor of variation into the independent components of a representation vector should make it both interpretable and simplify downstream prediction tasks [6, 7, 29, 35, 56, 58, 60, 74, 82, 87, 89, 90, 1, 28].

Previous work [54, 61] has alluded to a possible connection between the motivations of disentanglement and fair machine learning. Given the societal relevance of machine-learning driven decision processes, fairness has become a highly active field [4]. Assuming the existence of a complex causal graph with partially observed and potentially confounded observations [48], sensitive protected attributes (e.g. gender, race, etc) can leak undesired information into a classification task in different ways. For example, the inherent assumptions of the algorithm might cause discrimination towards protected groups, the data collection process might be biased or the causal graph itself might allow for unfairness because society is unfair [5, 11, 68, 73, 75, 83]. The goal of fair machine learning algorithms is to predict a target variable $\mathbf{y}$ through a classifier $\hat{\mathbf{y}}$ without being biased by some sensitive factors $\mathbf{s}$. The negative impact of $\mathbf{s}$ in terms of discrimination within the classification task can be quantified using a variety of fairness notions, such as demographic parity [10, 97], individual fairness [21], equalized odds or equal opportunity [34, 94], and concepts based on causal reasoning [48, 55].

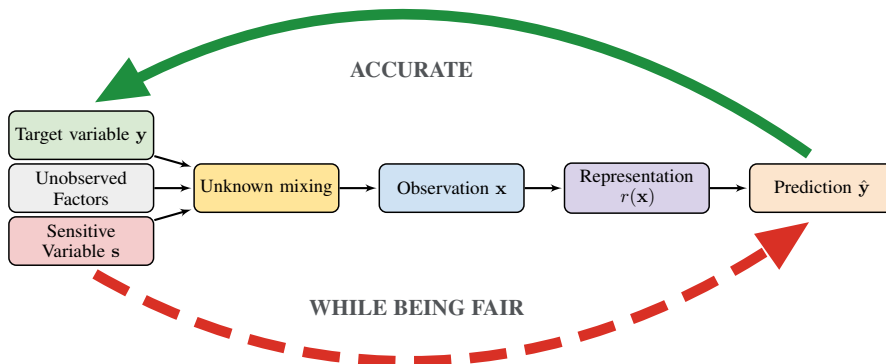

Figure 1: Causal graph and problem setting. We assume the observations $\mathbf{x}$ are manifestations of independent factors of variation. We aim at predicting the value of some factors of variation $\mathbf{y}$ without being influenced by the *unobserved* sensitive variable $\mathbf{s}$. Even though target and sensitive variable are in principle independent, they are entangled in the observations by an unknown mixing mechanism. Our goal for fair representation learning is to learn a good representation $r(\mathbf{x})$ so that any downstream classifier will be both accurate and fair. Note that the representation is learned without supervision and when training the classifier we do not observe and do not know which variables are sensitive.

In this paper, we investigate the downstream usefulness of disentangled representations through the lens of fairness. For this, we consider the standard setup of disentangled representation learning, in which observations are the result of an (unknown) mixing mechanism of independent ground-truth factors of variation as depicted in Figure 1. To evaluate the learned representations $r(\mathbf{x})$ of these observations, we assume that the set of ground-truth factors of variation include both a target factor $\mathbf{y}$, which we would like to predict from the learned representation, and an underlying sensitive factor $\mathbf{s}$, which we want to be fair to in the sense of demographic parity [10, 97], i.e. such that $p(\hat{\mathbf{y}} = y|\mathbf{s} = s_1) = p(\hat{\mathbf{y}} = y|\mathbf{s} = s_2) \; \forall y, s_1, s_2$. The key difference to prior work is that in this setting one never observes the sensitive variable $\mathbf{s}$ nor the other factors of variation except the target variable, which is itself only observed when learning the model for the downstream task. This setup is relevant when sensitive variables may not be recorded due to privacy reasons. Examples include learning general-purpose embeddings from a large number of images or building a world model based on video input of a robot.

Our key contributions can be summarized as follows:

- We motivate the setup of Figure 1 and discuss how general-purpose representations can lead to unfair predictions. In particular, we show theoretically that predictions can be unfair even if we use the Bayes optimal classifier and if the target variable and the sensitive variable are independent. Furthermore, we motivate why disentanglement in the representation may encourage fairness of the downstream prediction models.

- We evaluate the demographic parity of more than $90\,000$ downstream prediction models trained on more than $10\,000$ state-of-the-art disentangled representations on seven different data sets. Our results indicate that there are considerable dissimilarities between different representations in terms of fairness, indicating that the representation used matters.

- We relate the fairness of the representations to six different disentanglement scores of the same representations and find that disentanglement, in particular when measured using the DCI Disentanglement score [22], appears to be consistently correlated with increased fairness.

- We further investigate the relationship between fairness, the performance of the downstream models and the disentanglement scores. The fairness of the prediction also appears to be correlated to the accuracy of the downstream predictions, which is not surprising given that downstream accuracy is correlated with disentanglement.

**Roadmap:** In Section 2, we briefly review the state-of-the-art approaches to extract and evaluate disentangled representations. In Section 3, we highlight the role of the unknown mixing mechanism

on the fairness of the classification. In Section 4, we describe our experimental setup and empirical findings. In Section 5 we briefly review the literature on disentanglement and fair representation learning. In Section 6, we discuss our findings and their implications.

## 2   Background on learning disentangled representations

Consider the setup shown in Figure 1 where the observations $\mathbf{x}$ are caused by $k$ independent sources $z_1, \ldots, z_k$. The generative model takes the form of [71]:

$$p(\mathbf{x}, \mathbf{z}) = p(\mathbf{x} \mid \mathbf{z}) \prod_i p(z_i).$$

Informally, disentanglement learning treats the generative mechanisms as latent variables and aims at finding a representation $r(\mathbf{x})$ with independent components where a change in a dimension of $\mathbf{z}$ corresponds to a change in a dimension of $r(\mathbf{x})$ [6]. This intuitive definition can be formalized in a topological sense [35] and in the causality setting [87]. A large number of disentanglement scores measuring different aspects of disentangled representations have been proposed in recent years.

**Disentanglement scores.** The *BetaVAE* score [36] measures disentanglement by training a linear classifier to predict the index of a fixed factor of variation from the representation. The *FactorVAE* score [49] corrects a failure case of the BetaVAE score using a majority vote classifier on the relative variance of each dimension of $r(\mathbf{x})$ after intervening on $\mathbf{z}$. The *Mutual Information Gap (MIG)* [13] computes for each factor of variation the normalized gap on the top two entries in the matrix of pairwise mutual information between $\mathbf{z}$ and $r(\mathbf{x})$. The *Modularity* [79] measures if each dimension of $r(\mathbf{x})$ depends on at most one factor of variation using the matrix of pairwise mutual information between factors and representation dimensions. The Disentanglement metric of [22] (which we call *DCI Disentanglement* following [61]) is based on the entropy of the probability that a dimension of $r(\mathbf{x})$ is useful for predicting $\mathbf{z}$. This probability can be estimated from the feature importance of a random forest classifier. Finally, the *SAP score* [54] computes the average gap in the classification error of the two most predictive latent dimensions for each factor.

**Unsupervised methods.** State-of-the-art approaches for unsupervised disentanglement learning are based on representations learned by VAEs [51]. For the representation to be disentangled, the loss is enriched with a regularizer that encourages structure in the aggregate encoder distribution [2, 14, 13, 24, 36, 49, 65]. In causality, it is often argued that the true generative model is the simplest factorization of the distribution of the variables in the causal graph [74]. Under this hypothesis, $\beta$-VAE [36] and AnnealedVAE [9] limit the capacity of the VAE bottleneck so that it will be forced to learn disentangled representations. The Factor-VAE [49] and $\beta$-TCVAE [13] enforce that the aggregate posterior $q(\mathbf{z})$ is factorial by penalizing its total correlation. The DIP-VAE [54] and approach of [65] introduce a "disentanglement prior" for the aggregated posterior. We refer to Appendix B of [61] and Section 3 of [89] for a more detailed description of these regularizers.

## 3   The dangers of general purpose representations for fairness

Our goal in this paper is to understand how disentanglement impacts the fairness of general purpose representations. For this reason, we put ourselves in the simple setup of Figure 1 where we assume that the observations $\mathbf{x}$ depend on a set of independent ground-truth factors of variation through an unknown mixing mechanism. The key goal behind general purpose representations is to learn a vector valued function $r(\mathbf{x})$ that allows us to solve many downstream tasks that depend on the ground-truth factors of variation. From a representation learning perspective, a good representation should thus extract most of the information on the factors of variation [6], ideally in a way that enables easy learning from that representation, i.e., with few samples.

As one builds machine learning models for different tasks on top of such general purpose representations, it is not clear how the properties of the representations relate to the fairness of the predictions. In particular, for different downstream prediction tasks, there may be different sensitive variables that we would like to be fair to. This is modeled in our setting of Figure 1 by allowing one ground-truth factor of variation to be the target variable $\mathbf{y}$ and another one to be the sensitive variable $\mathbf{s}$.[1]

There are two key differences to prior setups in the fairness literature: First, we assume that one only observes the observations $\mathbf{x}$ when learning the representation $r(\mathbf{x})$ and the target variable $\mathbf{y}$ only when solving the downstream classification task. The sensitive variable $\mathbf{s}$ and the remaining ground-truth factors of variation are not observed. We argue that this is an interesting setting because for many large scale data sets labels may be scarce. Furthermore, if we can be fair with respect to unobserved but independent ground-truth factors of variation – for example by using disentangled representations, this might even allow us to avoid biases for sensitive factors that we are not aware of. The second difference is that we assume that the target variable $\mathbf{y}$ and the sensitive variable $\mathbf{s}$ are independent. While beyond the scope of this paper, it would be also interesting to study the setting where ground-truth factors of variations are dependent.

**Why can representations be unfair in this setting?** Despite the fact that the target variable $\mathbf{y}$ and the sensitive variable $\mathbf{s}$ are independent may seem like a overly restrictive assumption, we argue that even in this setting fairness is non-trivial to achieve. Since we only observe $\mathbf{x}$ or the learned representations $r(\mathbf{x})$, the target variable $\mathbf{y}$ and the sensitive variable $\mathbf{s}$ may be conditionally dependent. If we now train a prediction model based on $\mathbf{x}$ or $r(\mathbf{x})$, there is no guarantee that predictions will be fair with respect to $\mathbf{s}$.

There are additional considerations: first, the following theorem shows that the fairness notion of demographic parity may not be satisfied even if we find the optimal prediction model (i.e., $p(\hat{\mathbf{y}}|\mathbf{x}) = p(\mathbf{y}|\mathbf{x})$) on entangled representations (for example when the representations are the identity function, i.e. $r(\mathbf{x}) = \mathbf{x}$).

**Theorem 1.** *If $\mathbf{x}$ is entangled with $\mathbf{s}$ and $\mathbf{y}$, the use of a perfect classifier for $\hat{\mathbf{y}}$, i.e., $p(\hat{\mathbf{y}}|\mathbf{x}) = p(\mathbf{y}|\mathbf{x})$, does not imply demographic parity, i.e., $p(\hat{\mathbf{y}} = y|\mathbf{s} = s_1) = p(\hat{\mathbf{y}} = y|\mathbf{s} = s_2), \forall y, s_1, s_2$.*

The proof is provided in Appendix A. While this result provides a worst-case example, it should be interpreted with care. In particular, such instances may not allow for good and fair predictions regardless of the representations[2] and real world data may satisfy additional assumptions not satisfied by the provided counter example.

Second, the unknown mixing mechanism that relates $\mathbf{y}$, $\mathbf{s}$ to $\mathbf{x}$ may be highly complex and in practice the downstream learned prediction model will likely not be equal to the theoretically optimal prediction model $p(\hat{\mathbf{y}}|r(\mathbf{x}))$. As a result, the downstream prediction model may be unable to properly invert the unknown mixing mechanism and successfully separate $\mathbf{y}$ and $\mathbf{s}$, in particular as it may not be incentivized to do so. Finally, implicit biases and specific structures of the downstream model may interact and lead to different overall predictions for different sensitive groups in $\mathbf{s}$.

**Why might disentanglement help?** The key idea why disentanglement may help in this setting is that disentanglement promises to capture information about different generative factors in different latent dimensions. This limits the mutual information between different code dimensions and encourages the predictions to depend only on the latent dimensions corresponding to the target variable and not to the one corresponding to the sensitive ground-truth factor of variation. More formally, in the context of Theorem 1, consider a disentangled representation where the two factors of variations $\mathbf{s}$ and $\mathbf{y}$ are separated in independent components (say $r(\mathbf{x})_y$ only depends on $\mathbf{y}$ and $r(\mathbf{x})_s$ on $\mathbf{s}$). Then, the optimal classifier can learn to ignore the part of its input which is independent of $\mathbf{y}$ since $p(\hat{\mathbf{y}}|r(\mathbf{x})) = p(\mathbf{y}|r(\mathbf{x})) = p(\mathbf{y}|r(\mathbf{x})_y, r(\mathbf{x})_s) = p(\mathbf{y}|r(\mathbf{x})_y)$ as $\mathbf{y}$ is independent from $r(\mathbf{x})_s$. While such an optimal classifier on the representation $r(\mathbf{x})$ might be fairer than the optimal classifier on the observation $\mathbf{x}$, it may also have a lower prediction accuracy.

## 4 Do disentangled representations matter?

**Experimental conditions** We adopt the setup of [61], which offers the most extensive benchmark comparison of disentangled representations to date. Their analysis spans seven datasets: in four of them (*dSprites* [36], *Cars3D* [78], *SmallNORB* [59] and *Shapes3D* [49]), a deterministic function of the factors of variation is incorporated into the mixing process; they further introduce three additional variants of dSprites, *Noisy-dSprites*, *Color-dSprites*, and *Scream-dSprites*. In the latter datasets, the mixing mechanism contains a random component that takes the form of noisy pixels, random colors and structured backgrounds from the *scream* painting. Each of these seven datasets provides access to

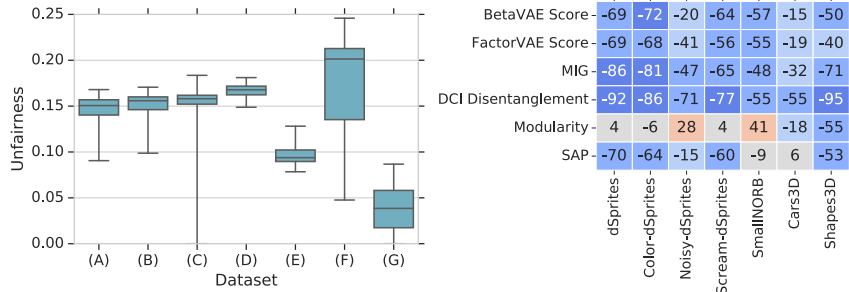

Figure 2: (Left) Distribution of unfairness for learned representations. Legend: dSprites = (A), Color-dSprites = (B), Noisy-dSprites = (C), Scream-dSprites = (D), SmallNORB = (E), Cars3D = (F), Shapes3D = (G). (Right) Rank correlation of unfairness and disentanglement scores on the various data sets.

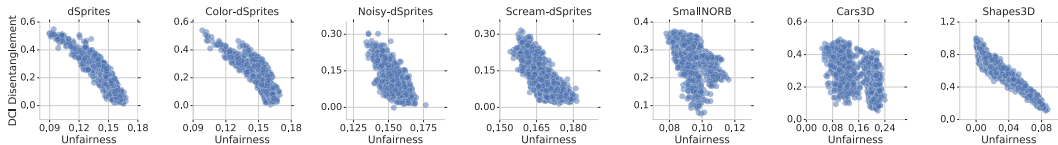

Figure 3: Unfairness of representations versus DCI Disentanglement on the different data sets.

the generative model for evaluation purposes. Our experimental pipeline works in three stages. First, we take the 12 600 pre-trained models of [61], which cover a large number of hyperparameters and random seeds for the most prominent approaches: $\beta$-VAE, AnnealedVAE, Factor-VAE, $\beta$-TCVAE, DIP-VAE-I and II. These methods are trained on the raw data without any supervision. Details on architecture, hyperparameter, implementation of the methods can be found in Appendices B, C, G, and H of [61]. In the second stage, we assume to observe a target variable $\mathbf{y}$ that we should predict from the representation while we do not observe the sensitive variable $\mathbf{s}$. For each trained model, we consider each possible pair of factors of variation as target and sensitive variables. For the prediction, we consider the same gradient boosting classifier [27] as in [61] which was trained on 10 000 labeled examples (subsequently denoted by GBT10000) and which achieves higher accuracy than the cross-validated logistic regression. In the third stage, we observe the values of all the factors of variations and have access to the whole generative model. With this we compute the disentanglement metrics and use the following score to measure the unfairness of the predictions

$$\texttt{unfairness}(\hat{\mathbf{y}}) = \frac{1}{|S|} \sum_s TV(p(\hat{\mathbf{y}}), p(\hat{\mathbf{y}} \mid \mathbf{s} = s)) \, \forall \, y$$

where $TV$ is the total variation. In other words, we compare the average total variation of the prediction after intervening on $\mathbf{s}$, thus directly measuring the violation of demographic parity. The reported unfairness score for each trained representation is the average unfairness of all downstream classification tasks we considered for that representation.

## 4.1 The unfairness of general purpose representations and the relation to dientanglement

In Figure 2 (left), we show the distribution of unfairness scores for different representations on different data sets. We clearly observe that learned representations can be unfair, even in the setting where the target variable and the sensitive variable are independent. In particular, the total variation can reach as much as $15\% - 25\%$ on five out of seven data sets. This confirms the importance of trying to find general-purpose representations that are less unfair.

We also observe in Figure 2 (left) that there is considerable spread in unfairness scores for different learned representations. This indicates that the specific representation used matters and that predictions with low unfairness can be achieved. To investigate whether disentanglement is a useful property to guarantee less unfair representations, we show rank correlation between a wide range of disentanglement scores and the unfairness score in Figure 2 (right). We observe that all disentanglement scores except Modularity appear to be consistently correlated with a lower unfairness

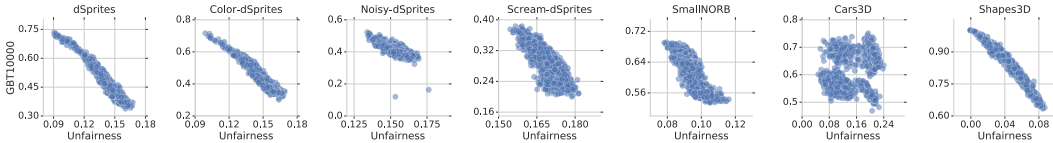

Figure 4: Unfairness of representations versus downstream accuracy on the different data sets.

Dataset = Color-dSprites (left)

| | Adj. BetaVAE Score | Adj. FactorVAE Score | Adj. MIG | Adj. DCI Disentanglement | Adj. Modularity | Adj. SAP |
|---|---|---|---|---|---|---|
| Adj. BetaVAE Score | 100 | 59 | 45 | 64 | 6 | 66 |
| Adj. FactorVAE Score | 59 | 100 | 32 | 52 | 1 | 41 |
| Adj. MIG | 45 | 32 | 100 | 69 | -5 | 57 |
| Adj. DCI Disentanglement | 64 | 52 | 69 | 100 | -18 | 66 |
| Adj. Modularity | 6 | 1 | -5 | -18 | 100 | -7 |
| Adj. SAP | 66 | 41 | 57 | 66 | -7 | 100 |

Dataset = Color-dSprites (right)

| | Adj. BetaVAE Score | Adj. FactorVAE Score | Adj. MIG | Adj. DCI Disentanglement | Adj. Modularity | Adj. SAP |
|---|---|---|---|---|---|---|
| BetaVAE Score | 51 | 38 | 27 | 41 | 5 | 43 |
| FactorVAE Score | 37 | 66 | 21 | 37 | -2 | 32 |
| MIG | 22 | 22 | 45 | 37 | -0 | 37 |
| DCI Disentanglement | 28 | 28 | 30 | 47 | -9 | 38 |
| Modularity | 5 | 2 | -3 | -15 | 85 | -7 |
| SAP | 36 | 26 | 34 | 43 | -7 | 66 |

Figure 5: Rank correlation between the adjusted disentanglement scores (left) and between original scores and the adjusted version (right).

score for all data sets. While the considered disentanglement metrics (except Modularity) have been found to be correlated (see [61]), we observe significant differences in between scores: Figure 2 (right) indicates that DCI Disentanglement is correlated the most followed by the Mutual Information Gap, the BetaVAE score, the FactorVAE score, the SAP score and finally Modularity. The strong correlation of DCI Disentanglement is confirmed by Figure 3 where we plot the Unfairness score against the DCI Disentanglement score for each model. Again, we observe that the large gap in unfairness seem to be related to differences in the representation. We show the corresponding plots for all metrics in Figure 9 in the Appendix.

These results provide an encouraging case for disentanglement being helpful in finding fairer representations. However, they should be interpreted with care: Even though we have considered a diverse set of methods and disentangled representations, the computed correlation scores depend on the distribution of considered models. If one were to consider an entirely different set of methods, hyperparameters and corresponding representations, the observed relationship may differ.

## 4.2 Adjusting for downstream performance

Prior work [61] has observed that disentanglement metrics are correlated with how well ground-truth factors of variations can be predicted from the representation using gradient boosted trees. It is thus not surprising that the unfairness of a representation is also consistently correlated to the average accuracy of a gradient boosted trees classifier using $10\,000$ samples (see Figure 4). In this section, we investigate whether disentanglement is also correlated with a higher fairness if we compare representations with the same accuracy as measured by GBT10000 scores. Given two representations with the same downstream performance, is the more disentangled one also more fair? The key challenge is that for a given representation there may not be other ones with exactly the same downstream performance.

For this, we adjust all the disentanglement scores and the unfairness score for the effect of downstream performance. We use a k-nearest neighbors regression from Scikit-learn [72] to predict, for any model, each disentanglement score and the unfairness from its five nearest neighbor in terms of GBT10000 (which we write as $N(\text{GBT10000})$). This can be seen as a one-dimensional non-parametric estimate of the disentanglement score (or fairness score) based on the GBT10000 score. The adjusted metric is computed as the residual score after the average score of the neighbors is subtracted, namely

$$\texttt{Adj. Metric} = \texttt{Metric} - \frac{1}{5} \sum_{i \in N(\text{GBT10000})} \texttt{Metric}_i$$

Intuitively, the adjusted metrics measure how much more disentangled (fairer) a given representation is compared to an average representation with the same downstream performance.

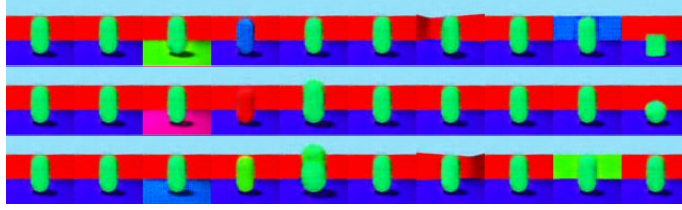

Figure 6: Latent traversals (each column corresponds to a different latent variable being varied) on Shapes3D for the model with best adjusted MIG.

In Figure 5 (left), we observe that the rank correlation between the adjusted disentanglement scores (except Modularity) on Color-dSprites is consistently positive. This indicates that the adjusted scores do measure a similar property of the representation even when adjusted for performance. This result is consistent across data sets (Figure 10 of the Appendix). The only exception appears to be SmallNORB, where the adjusted DCI Disentanglement, MIG and SAP score correlate with each other but do not correlate well with the BetaVAE and FactorVAE score (which only correlate with each other). On Shapes3D we observe a similar result, but the correlation between the two groups of scores is stronger than on SmallNORB. Similarly, Figure 5 (right) shows the rank correlation between the disentanglement metrics and their adjusted versions. As expected, we observe that there still is a significant positive correlation. This indicates the adjusted scores still capture a significant part of the unadjusted score. We observe in Figure 11 of the Appendix that this result appears to be consistent across the different data sets, again with the exception of SmallNORB. As a sanity check, we finally confirm by visual inspection that the adjusted metrics still measure disentanglement. In Figure 6, we plot latent traversals for the model with highest adjusted MIG score on Shapes3D and observe that the model appears well disentangled.

|                        | dSprites | Color-dSprites | Noisy-dSprites | Scream-dSprites | SmallNORB | Cars3D | Shapes3D |
|------------------------|------|------|------|------|------|------|------|
| Adj. BetaVAE Score     | -18  | -26  | -19  | -10  | -20  | -12  | 25   |
| Adj. FactorVAE Score   | -16  | -26  | -39  | -12  | -23  | -15  | 13   |
| Adj. MIG               | -27  | -37  | 0    | -12  | 19   | -10  | -23  |
| Adj. DCI Disentanglement | -33 | -45 | -13  | -13  | 20   | -28  | -41  |
| Adj. Modularity        | 9    | 15   | 24   | 1    | 16   | -13  | -3   |
| Adj. SAP               | -17  | -27  | -4   | -13  | 16   | 1    | 3    |

Figure 7: Rank correlation of unfairness and disentanglement scores on the various data sets (left). Rank correlation of adjusted unfairness and adjusted disentanglement scores on the various data sets (right).

Finally, Figure 7 shows the rank correlation between the adjusted disentanglement scores and the adjusted fairness score for each of the data sets. Overall, we observe that higher disentanglement still seems to be correlated with an increased fairness, even when accounting for downstream performance. Exceptions appear to be the adjusted Modularity score, the adjusted BetaVAE and the FactorVAE score on Shapes3D, and the adjusted MIG, DCI Disentanglement, Modularity and SAP on SmallNORB. As expected, the correlations appear to be weaker than for the unadjusted scores (see Figure 2 (right)) but we still observe some residual correlation.

**How do we identify fair models?** In this section, we observed that disentangled representations allow to train fairer classifiers, regardless of their accuracy. This leaves us with the question of how can we find fair representations? [61] showed that without access to supervision or inductive biases, disentangled representations cannot be identified. However, existing methods heavily rely on inductive biases such as architecture, hyperparameter choices, mean-field assumptions, and smoothness induced through randomness [65, 80, 86]. In practice, training a large number of models with different losses and hyperparameters will result in a large number of different representations, some of which might be more disentangled than others as can be seen for example in Figure 3. From Theorem 1, we know that optimizing for accuracy on a fixed representation does not guarantee to learn a fair classifier as the demographic parity theoretically depends on the representation when the sensitive variable is not observed.

When we fix a classification algorithm, in our case GBT10000, and we train it over a variety of representations with different degrees of disentanglement we obtain both different degrees of fairness and downstream performance. If the disentanglement of the representation is the *only* confounder between the performance of the classifier and its fairness as depicted in Figure 8, the classification accuracy may be used as a proxy for fairness. To test whether this holds in practice, we perform the

following experiment. We sample a data set, a seed for the unsupervised disentanglement models and among the factors of variations we sample one to be $\mathbf{y}$ and one to be $\mathbf{s}$. Then, we train a classifier predicting $\mathbf{y}$ from $r(\mathbf{x})$ using all the models trained on that data set on the specific seed. We compare the unfairness of the classifier achieving highest prediction accuracy on $\mathbf{y}$ with a randomly chosen classifier from the ones we trained. We observe that the classifier selected using test accuracy is also fairer 84.2% of the times. We remark that this result explicitly make use of a large amount of representations of different quality on which we train the same classification algorithm. Under the assumption of Figure 8, the disentanglement of the representation is the only difference explaining different predictions, the best performing classifier is also more fair than one trained on a different representation. Since disentanglement is likely not the only confounder, model selection based on downstream performance is not guaranteed to always be fairer than random model selection.

## 5 Related Work

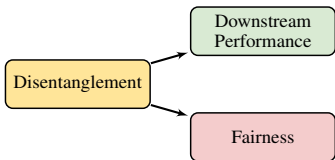

Figure 8: If disentanglement is a causal parent of downstream performance and fairness and there are no hidden confounders, then the former can be used as a proxy for the latter.

Ideas related to disentangling the factors of variations have a long tradition in machine learning, dating back to the non-linear ICA literature [17, 3, 46, 42, 43, 44, 32]. Disentangling pose from content and content from motion are also classical computer vision problems that have been tackled with various degrees of supervision and inductive bias [92, 93, 40, 25, 18, 31, 42]. In this paper, we intend disentanglement in the sense of [6, 87, 35, 61]. [61] recently proved that without access to supervision or inductive biases, disentanglement learning is impossible as disentangled models cannot be identified. In this paper, we evaluate the representation using the supervised downstream task where both target and sensitive variables are observed. Semi-supervised variants have been extensively studied during the years. [77, 15, 66, 70, 50, 52, 1] assume partially observed factors of variation that should be disentangled from the other unobserved ones. Weaker forms of supervision like relational information or additional assumptions on the effect of the factors of variation were also studied [39, 16, 47, 31, 91, 26, 19, 41, 93, 62, 53, 81, 8] and applied in the sequential data and reinforcement learning settings [88, 85, 57, 69, 37, 38]. Overall, the disentanglement literature is interested in isolating the effect of every factor of variation regardless of how the representation should be used downstream.

On the fairness perspective, representation learning has been used as a mean to separate the detrimental effects that labeled sensitive factors could have on the classification task [67, 34]. We remark that this setup is different from what we consider in this paper, as we do not assume access to any labeled information when learning a representation. In particular, we do not assume to know what the downstream task will be and what are the sensible variables (if any). [21, 95] introduce the idea that a fair representation should preserve all information about the individual's attributes except for the membership to protected groups. In practice, [63] extends the VAE objective with a Maximum Mean Discrepancy [33] to ensure independence between the latent representation and the sensitive factors. [12] introduces the idea of data pre-processing as a tool to control for downstream discrimination. The authors of [84] instead propose an information-theoretic approach in which the mutual information between the data and the representation is maximized, while the one between the sensitive attributes and the representation is minimized. Furthermore, there are several approaches that employ adversarial [30] training to avoid information leakage between the sensitive attributes and the representation [23, 64, 96]. Finally, representation learning has recently proved to be useful in counterfactual fairness [55, 45].

## 6 Conclusion

In this paper, we observe the first empirical evidence that disentanglement might prove beneficial to learn fair representations, providing evidence supporting the conjectures of [61, 54]. We show that general purpose representations can lead to substantial unfairness, even in the setting where both the sensitive variable and target variable are independent and one only has access to observations that depend on both of them. Yet, the choice of representation appears to be crucial as we find that that increased disentanglement of a representation is consistently correlated with increased fairness on

downstream prediction tasks across a wide range of representations and data sets. Furthermore, we discuss the relationship between fairness, downstream accuracy and disentanglement and find evidence that the correlation between disentanglement metrics and the unfairness of the downstream prediction tasks appears to also hold if one accounts for the downstream accuracy. We believe that these results serve as a motivation for further investigation on the practical benefits of disentangled representations, especially in the context of fairness. Finally, we argue that fairness should be among the desired properties of general purpose representation learning beyond VAEs [20, 76]. As we highlighted in this paper, it appears possible to learn representations that are both useful, interpretable and fairer. Progress on this problem could allow machine-learning driven decision making to be both better and fairer.

## Acknowledgements

The authors thank Sylvain Gelly and Niki Kilbertus for helpful discussions and comments. Francesco Locatello is supported by the Max Planck ETH Center for Learning Systems, by an ETH core grant (to Gunnar Rätsch), and by a Google Ph.D. Fellowship. This work was partially done while Francesco Locatello was at Google Research Zurich. Gabriele Abbati acknowledges funding from Google Deepmind and the University of Oxford. Tom Rainforth is supported in part by the European Research Council under the European Union's Seventh Framework Programme (FP7/2007–2013) / ERC grant agreement no. 617071 and in part by EPSRC funding under grant EP/P026753/1.

## Footnotes

[1]Please see Section 4.1 for how this is done in the experiments.

[2]In this case, even properties of representations such as disentanglement may not help.

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
