[Reviews · NeurIPS 2019]

Reviewer 1



Response read. I appreciate the author's commitment to making the paper clearer. The authors addressed my main concerns. I will upgrade to an 8. ---------------- **Summary** The authors attempt to answer the question: Do disentangled representations help with fairness? They do so by training 12,600 disentangling models and testing to what extent fair prediction can be done on top of these representations. They show experimentally that disentangling not only correlates to fairness, but even so when they “adjust” for accuracy” This leads them to surmise that maybe disentangling is a common cause for accuracy and fairness and thus maybe if it is the only common cause then when dealing with disentangled models, they could use accuracy as a proxy for fairness. **Strengths** * very well-written and clear for the most part * nice concise summary of metrics, disentangling as a whole * all claims backed up with thorough large scale experiments (12,600 models) * experimental exploration of disentangling and fairness at this scale seems novel to me * could be very significant to fairness community if disentangling is the missing piece to ensure fairness in classifiers **Weakness** * The title is elegant, but it seems a bit general/all-encompassing. Maybe mention fairness in the title? Aren’t there other benefits of disentangled representations not covered here (sample efficiency, transfer/generalization, etc.) ? * Section 4.2 is quite clear and well-written for the most part, but there are a couple confusing parts. For me, in the adjusted score part it was a bit confusing to understand how subtracting the disentangling score from the nearest neighbors (in terms of classification accuracy) achieves the effect of “removing” the effect of accuracy. * In the "How do we identify fair models?" section, I am a bit confused at how the chain of logic concluded that classification accuracy could be a proxy for fairness. Perhaps a causal graph between three nodes (disentangling score, GBT10000 accuracy, fairness) could be drawn and explained. I would be curious to see if my understanding is correct: ’’’’ accuracy does not always lead to fairness (thm 1), disentangling correlates with accuracy (Locatello, et al. ), disentangling correlates with fairness (section 4.1), disentangling correlates with fairness even if we adjust for accuracy (section 4.2), accuracy correlates with fairness in disentangling models (figure 4), so then disentangling is maybe a “common cause” between accuracy and fairness aka fairness <- disentangling -> accuracy. So then if this is true (if disentangling is the only confounder), the model with highest accuracy should be the most disentangled, which means the fairest?’’’’ Overall, a nice thoughtful paper with thorough experiments and even though I don't know much about fairness, it seems it could have some significance for the fairness community

Reviewer 2



Originality The idea of improving the fairness of predictions via disentangling looks novel to me. Quality Most of the claims in this paper are supported by theorems and experiments. But it looks like there are some issues as I presented in Section 1. Clarity In general, this paper is well-organized and not difficult to follow. I am not able to understand GBT 10000 scores and adj. metric. The authors might need to briefly introduce "gradient boosted trees classifier" and the motivation of computing adj. metric. Significance I believe this paper will have reasonable contributions if the authors can fix the issues I have mentioned above.

Reviewer 3



- Predict a target variable based on representations. - Theory suggests disentanglement doesn't guarantee fairness, but empirical results show a correlation between fairness and disentanglement. - Fairness here is defined as having a prediction not depend on a sensitive factor s. - The introduction could be a bit clearer about the definition of fairness that will be used, given the technical nature of the paper. More particularly, what justifies the demographic parity definition that is used throughout the rest of the paper? - In Figure 1, assumes a causal graph where the sensitive and target variables are independent, but the observations are generated from a complicated and unknown mixing function. - Intuitively I can think of a simple case where having entangled latent factors can "drag" along irrelevant factors which are correlated with the sensitive variable. - Several VAE variants try to improve disentanglement. - May be useful to also be fair with respect to unobserved variables. - Experiments show that disentanglement is generally correlated with fairness (Cars3D seems to be an exception) -- This paper presents an interesting empirical analysis relating fairness to disentanglement of learned representations.

[Author Response · NeurIPS 2019]

We thank the reviewers for their consideration of our paper and for their feedback. The consensus appears to be that this is a generally well written paper, exploring a *"potentially impressive"* (R2), *"useful"* (R1, R3) and *"novel"* (R1, R2) connection between fairness and disentanglement with claims backed up by substantial empirical evidence (R1, R2, R3). There appears to be one major concern and 4 minor questions/suggestions, which we kindly address below.

**Main concern of R2: use of ground-truth factors to compute disentanglement scores**

The main concern of R2 seems to be that the paper relies on *"disentanglement scores, which are computed based on ground-truth factors of variations"* but that these factors *"are generally not available in practice"*. In our view, there appears to be a misunderstanding with regards to the exact goals and contributions of this paper:

- **Motivation: usefulness of disentanglement notions vs new method.** The main goal of this paper is to *evaluate the usefulness* of disentangled representations (in terms of leading to fair predictions), rather than proposing new methods for *learning* disentangled representations. This distinction is critical: While we do train various methods for the purpose of this study, we do not make any assumptions about the *feasibility* of (and *methods* for) learning disentangled representations in the absence of ground-truth factors, and our results have implications for the supervised, semi-supervised, and unsupervised settings. This paper provides evidence that, if one were to find a reliable method for disentanglement (in terms of current metrics), predictions on representations of such an approach would be more fair.

- **Relevance: validate the motivation of >15 recent ML papers.** Recently, numerous papers have been concerned with learning disentangled representations [7, 8, 12, 17, 18, 20, 31, 32, 33, 34, 43, 50, 57, 61, 66, 74, 76, 82, 84]. The key motivation (but also assumption) of these works is that current notions of disentanglement (MIG, DCI, *etc.*) are desirable. Until now there has been little empirical evidence verifying this. In particular, the study of Locatello et al., ICML 2019 was inconclusive in this regard, see their Section 5.5. This paper fills this gap by investigating whether disentanglement is useful for improving fairness of downstream predictions. Our results are novel and provide motivation for further research into (a) disentanglement methods and (b) their application for ML fairness.

- **We present a heuristic to select fair representations.** As described in Section 4.2, as a by-product of our investigation, we noticed that downstream predictive performance may be used as a way to select fair representations among the representations we trained using SOTA unsupervised disentanglement methods. We argue that this might be an interesting heuristic as it only requires labels for the single downstream prediction task but not for the representation learning.

**Other concerns:**

**R1-R2:** *Motivation for adjusted metrics in Section 4.2.*

We compute the adjusted metrics to answer the question **"Given two representations with the same downstream performance, is the more disentangled one also more fair?"**. The key difficulty is that for a given representation there may not be other ones with exactly the same downstream performance. Hence, we compute these adjusted metrics which intuitively measure how much fairer (more disentangled) a representation is compared to an average representation with the same downstream performance. To compute the average fairness (disentanglement) for a given downstream performance, we use a nearest neighbor regressor as a robust non-parametric 1D regression model.

**R1:** *Chain of arguments in "How do we identify fair models?" not fully clear. + Is the argument by R1 correct?*

**The understanding of the reviewer is correct.** We will add the suggested graph "fairness <- disentangling -> accuracy" and revise the manuscript such that this argument is explained more concisely.

**R2:** *Theorem 1.*

**Entangled representations affect the fairness of downstream classifiers.** The theorem proves a point: the Bayes optimal classifier does not imply fairness on an entangled representation. The proof is by counterexample. We further support this claim with Figure 2, showing that this is a practical issue that arises on trained classifiers and not simply a theoretical corner case. If we can disentangle target and sensitive variables then the classifier will be fair, see lines 145-155. In the particular counterexample of the theorem, there is a trade-off between fairness and accuracy as is common in the fairness literature. This is discussed in lines 135-138 and in the footnote.

**R3:** *Demographic parity.*

**We will add more motivation and examples,** in particular relating to previous work (e.g. [9, 92]) and to relevant settings, for example to settings where sensitive variables may not be recorded due to privacy reasons.

*Clarity, title, and other comments regarding the presentations of our results.*

**We thank the reviewers for their suggestions to improve the presentation.** We will use their feedback to revise the manuscript. In particular, as suggested, we will change the title to "On the Fairness of Disentangled Representations", improve the discussion on section 4.2 and add a discussion on non-VAE based methods such as ALI. By gradient boosted trees (GBT10000) we mean gradient boosting of decision trees [Friedman, 1999] trained on 10000 examples.

[Meta-Review · NeurIPS 2019]

The authors evaluate the potential benefits of disentangled representations with respect to fairness. The paper offers extensive empirical results that suggest that disentanglement metrics correlate with measures of fairness. This is an important finding insofar as it motivates continued research toward disentanglement. Reviewers have highlighted some issues with presentation (e.g., the title might be edited to explicitly mention fairness) which should be addressed in future versions of the manuscript.